# Telescoping Density-Ratio Estimation

**Benjamin Rhodes**
School of Informatics
University of Edinburgh
ben.rhodes@ed.ac.uk

**Kai Xu**
School of Informatics
University of Edinburgh
kai.xu@ed.ac.uk

**Michael U. Gutmann**
School of Informatics
University of Edinburgh
michael.gutmann@ed.ac.uk

## Abstract

Density-ratio estimation via classification is a cornerstone of unsupervised learning. It has provided the foundation for state-of-the-art methods in representation learning and generative modelling, with the number of use-cases continuing to proliferate. However, it suffers from a critical limitation: it fails to accurately estimate ratios $p/q$ for which the two densities differ significantly. Empirically, we find this occurs whenever the KL divergence between $p$ and $q$ exceeds tens of nats. To resolve this limitation, we introduce a new framework, telescoping density-ratio estimation (TRE), that enables the estimation of ratios between highly dissimilar densities in high-dimensional spaces. Our experiments demonstrate that TRE can yield substantial improvements over existing single-ratio methods for mutual information estimation, representation learning and energy-based modelling.

## 1 Introduction

Unsupervised learning via density-ratio estimation is a powerful paradigm in machine learning [60] that continues to be a source of major progress in the field. It consists of estimating the ratio $p/q$ from their samples without separately estimating the numerator and denominator. A common way to achieve this is to train a neural network classifier to distinguish between the two sets of samples, since for many loss functions the ratio $p/q$ can be extracted from the optimal classifier [60, 21, 41]. This discriminative approach has been leveraged in diverse areas such as covariate shift adaptation [59, 63], energy-based modelling [22, 4, 53, 64, 36, 19], generative adversarial networks [15, 47, 43], bias correction for generative models [20, 18], likelihood-free inference [50, 62, 8, 13], mutual-information estimation [2], representation learning [29, 30, 48, 25, 27], Bayesian experimental design [33, 34] and off-policy reward estimation in reinforcement learning [39]. Across this diverse set of applications, density-ratio based methods have consistently yielded state-of-the-art results.

Despite the successes of discriminative density-ratio estimation, many existing loss functions share a severe limitation. Whenever the 'gap' between $p$ and $q$ is large, the classifier can obtain almost perfect accuracy with a relatively poor estimate of the density ratio. We refer to this failure mode as the *density-chasm problem*—see Figure 1a for an illustration. We observe empirically that the density-chasm problem manifests whenever the KL-divergence $D_{KL}(p \parallel q)$ exceeds $\sim 20$ nats[1]. This observation accords with recent findings in the mutual information literature regarding the limitations of density-ratio based estimators of the KL [40, 52, 57]. In high dimensions, it can easily occur that two densities $p$ and $q$ will have a KL-divergence measuring in the hundreds of nats, and so the ratio may be virtually intractable to estimate with existing techniques.

In this paper, we propose a new framework for estimating density-ratios that can overcome the density-chasm problem. Our solution uses a 'divide-and-conquer' strategy composed of two steps. The first step is to gradually transport samples from $p$ to samples from $q$, creating a chain of intermediate datasets. We then estimate the density-ratio between consecutive datasets along this

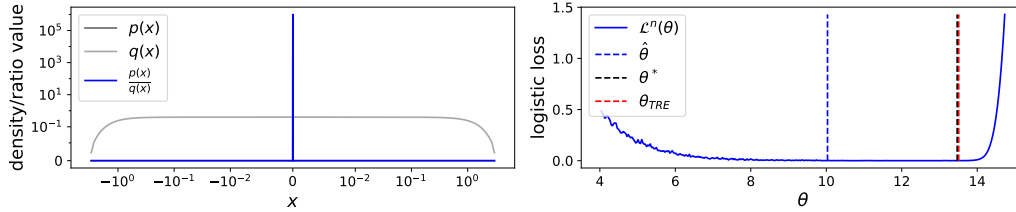

(a) Density-ratio estimation between an extremely peaked Gaussian $p$ ($\sigma = 10^{-6}$) and a broad Gaussian $q$ ($\sigma = 1$) using a single-parameter quadratic classifier (as detailed in section 4.1). **Left**: A log-log scale plot of the densities and their ratio. Note that p(x) is not visible, since the ratio overlaps it. **Right**: the solid blue line is the finite-sample logistic loss (Eq. 2) for 10,000 samples. Despite the large sample size, the minimiser (dotted blue line) is far from optimal (dotted black line). The dotted red line is the newly introduced TRE solution, which almost perfectly overlaps with the dotted black line.

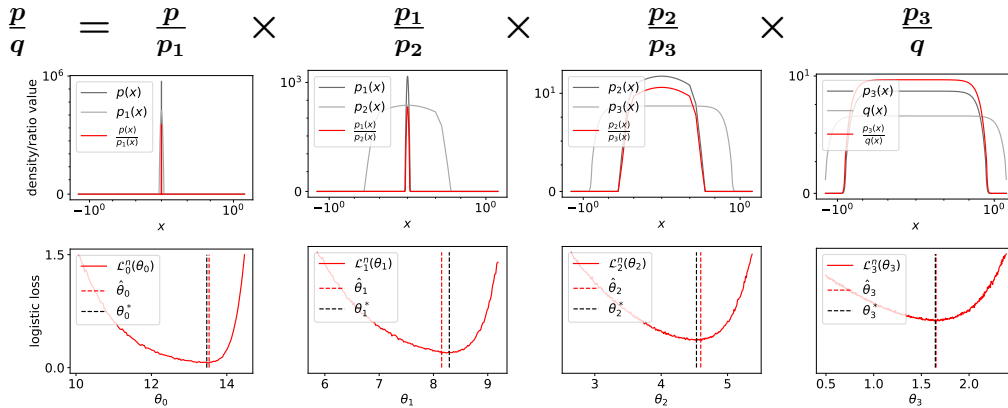

(b) Telescoping density-ratio estimation applied to the problem in (a), using the same 10,000 samples from $p$ and $q$. **Top row**: a collection of ratios, where $p_1$, $p_2$ and $p_3$ are constructed by deterministically interpolating between samples from $p$ and $q$. **Bottom row**: the logistic loss function for each ratio estimation problem. Observe that the finite-sample minimisers of each objective (red dotted lines) are either close to or exactly overlapping their optima (black dotted lines). After estimating each ratio, we then combine them by taking their product.

Figure 1: Illustration of standard density-ratio estimation vs. telescoping density-ratio estimation.

chain, as illustrated in the top row of Figure 1b. Unlike the original ratio $p/q$, these 'chained ratios' can be accurately estimated via classification (see bottom row). Finally, we combine the chained ratios via a telescoping product to obtain an estimate of the original density-ratio $p/q$. Thus, we refer to the method as Telescoping density-Ratio Estimation (TRE).

We empirically demonstrate that TRE can accurately estimate density-ratios using deep neural networks on high-dimensional problems, significantly outperforming existing single-ratio methods. We show this for two important applications: representation learning via mutual information (MI) estimation and the learning of energy-based models (EBMs).

In the context of mutual information estimation, we show that TRE can accurately estimate large MI values of 30+ nats, which is recognised to be an outstanding problem in the literature [52]. However, obtaining accurate MI estimates is often not our *sole* objective; we also care about learning representations from e.g. audio or image data that are useful for downstream tasks such as classification or clustering. To this end, our experimental results for representation learning confirm that TRE offers substantial gains over a range of existing single-ratio baselines.

In the context of energy-based modelling, we show that TRE can be viewed as an extension of noise-contrastive estimation [22] that more efficiently scales to high-dimensional data. Whilst energy-based modelling has been a topic of interest in the machine learning community for some time [56], there has been a recent surge of interest, with a wave of new methods for learning deep EBMs in high dimensions [10, 6, 58, 38, 17, 68]. These methods have shown promising results for image and 3D shape synthesis [66], hybrid modelling [16], and modelling of exchangeable data [67].

However, many of these methods result in expensive/challenging optimisation problems, since they rely on approximate Markov chain Monte Carlo (MCMC) sampling during learning [10, 16, 68], or on adversarial optimisation [6, 17, 68]. In contrast, TRE requires no MCMC during learning and uses a well-defined, non-adversarial, objective function. Moreover, as we show in our mutual information experiments, TRE is applicable to discrete data, whereas all other recent EBM methods only work for continuous random variables. Applicability to discrete data makes TRE especially promising for domains such as natural language processing, where noise-contrastive estimation has been widely used [42, 35, 1].

## 2 Discriminative ratio estimation and the density-chasm problem

Suppose $p$ and $q$ are two densities for which we have samples, and that $q(\mathbf{x}) > 0$ whenever $p(\mathbf{x}) > 0$. We can estimate the density-ratio $r(\mathbf{x}) = p(\mathbf{x})/q(\mathbf{x})$ by training a classifier to distinguish samples from $p$ and $q$ [23, 60, 22]. There are many choices for the loss function of the classifier [60, 51, 21, 41, 52], but in this paper we concentrate on the widely used logistic loss

$$\mathcal{L}(\boldsymbol{\theta}) = -\mathbb{E}_{\mathbf{x}_1 \sim p} \log \left( \frac{r(\mathbf{x}_1; \boldsymbol{\theta})}{1 + r(\mathbf{x}_1; \boldsymbol{\theta})} \right) - \mathbb{E}_{\mathbf{x}_2 \sim q} \log \left( \frac{1}{1 + r(\mathbf{x}_2; \boldsymbol{\theta})} \right), \tag{1}$$

where $r(\mathbf{x}; \boldsymbol{\theta})$ is a non-negative ratio estimating model. To enforce non-negativity, $r$ is typically expressed as the exponential of an unconstrained function such as a neural network. For a correctly specified model, the minimiser of this loss, $\boldsymbol{\theta}^*$, satisfies $r(\mathbf{x}; \boldsymbol{\theta}^*) = p(\mathbf{x})/q(\mathbf{x})$, without needing any normalisation constraints [22]. Other classification losses do not always have this self-normalising property, but only yield an estimate proportional to the true ratio—see e.g. [52].

**The density-chasm problem**

We experimentally find that density-ratio estimation via classification typically works well when $p$ and $q$ are 'close' e.g. the KL divergence between them is less than $\sim 20$ nats. However, for sufficiently large gaps, which we refer to as *density-chasms*, the ratio estimator is often severely inaccurate. This raises the obvious question: what is the cause of such inaccuracy?

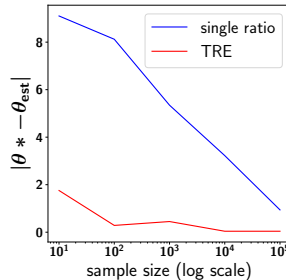

Figure 2: Sample efficiency curves for the experiment in Figure 1. Single ratio estimation can be extremely sample-inefficient.

There are many possible sources of error: the use of misspecified models, imperfect optimisation algorithms, and inaccuracy stemming from Monte Carlo approximations of the expectations in (1). We argue that this mundane final point—Monte Carlo error due to finite sample size—is actually sufficient for inducing the density-chasm problem. Figure 1a depicts a toy problem for which the model is well-specified, and because it is 1-dimensional (w.r.t. $\theta$), optimisation is straightforward using grid-search. And yet, if we use a sample size of $n = 10,000$ and minimise the finite-sample loss

$$\mathcal{L}^n(\theta) = \sum_{i=1}^{n} -\log \left( \frac{r(x_1^i; \theta)}{1 + r(x_1^i; \theta)} \right) - \log \left( \frac{1}{1 + r(x_2^i; \theta)} \right), \qquad x_1^i \sim p, \; x_2^i \sim q, \tag{2}$$

we obtain an estimate $\hat{\theta}$ that is far from the asymptotic minimiser $\theta^* = \arg \min \mathcal{L}(\theta)$. Repeating this same experiment for different sample sizes, we can empirically measure the method's sample efficiency, which is plotted as the blue curve in Figure 2. For the regime plotted, we see that an exponential increase in sample size only yields a linear decrease in estimation error. This empirical result is concordant with theoretical findings that density-ratio based lower bounds on KL divergences are only tight for sample sizes exponential in the the number of nats [40].

Whilst we focus on the logistic loss, we believe the density chasm problem is a broader phenomenon. As shown in the appendix, the issues identified in Figure 1 and the sample inefficiency seen in Figure 2 also occur for other commonly used discriminative loss functions.

Thus, when faced with the density-chasm problem, simply increasing the sample size is a highly inefficient solution and not always possible in practice. This begs the question: is there a more intelligent way of using a fixed set of samples from $p$ and $q$ to estimate the ratio?

# 3 Telescoping density-ratio estimation

We introduce a new framework for estimating density-ratios $p/q$ that can overcome the density-chasm problem in a *sample-efficient* manner. Intuitively, the density-chasm problem arises whenever classifying between $p$ and $q$ is 'too easy'. This suggests that it may be fruitful to decompose the task into a collection of harder sub-tasks.

For convenience, we make the notational switch $p \equiv p_0$, $q \equiv p_m$ (which we will keep going forward), and expand the ratio via a telescoping product

$$\frac{p_0(\mathbf{x})}{p_m(\mathbf{x})} = \frac{p_0(\mathbf{x})}{p_1(\mathbf{x})} \frac{p_1(\mathbf{x})}{p_2(\mathbf{x})} \cdots \frac{p_{m-2}(\mathbf{x})}{p_{m-1}(\mathbf{x})} \frac{p_{m-1}(\mathbf{x})}{p_m(\mathbf{x})}, \tag{3}$$

where, ideally, each $p_k$ is chosen such that a classifier cannot easily distinguish it from its two neighbouring densities. Instead of attempting to build one large 'bridge' (i.e. density-ratio) across the density-chasm, we propose to build many small bridges between intermediate 'waymark' distributions. The two key components of the method are therefore:

1. **Waymark creation.** We require a method for *gradually* transporting samples $\{\mathbf{x}_0^1, \ldots, \mathbf{x}_0^n\}$ from $p_0$ to samples $\{\mathbf{x}_m^1, \ldots, \mathbf{x}_m^n\}$ from $p_m$. At each step in the transportation, we obtain a new dataset $\{\mathbf{x}_k^1, \ldots, \mathbf{x}_k^n\}$ where $k \in \{0, \ldots m\}$. Each intermediate dataset can be thought of as samples from an implicit distribution $p_k$, which we refer to as a *waymark* distribution.

2. **Bridge-building:** A method for learning a set of parametrised density-ratios between consecutive pairs of waymarks $r_k(\mathbf{x}; \boldsymbol{\theta}_k) \approx p_k(\mathbf{x})/p_{k+1}(\mathbf{x})$ for $k = 0, \ldots, m-1$, where each bridge $r_k$ is a non-negative function. We refer to these ratio estimating models as *bridges*. Note that the parameters of the bridges, $\{\boldsymbol{\theta}_k\}_{k=0}^{m-1}$, can be totally independent or they can be partially shared.

An estimate of the original ratio is then given by the product of the bridges

$$r(\mathbf{x}; \boldsymbol{\theta}) = \prod_{k=0}^{m-1} r_k(\mathbf{x}; \boldsymbol{\theta}_k) \approx \prod_{k=0}^{m-1} \frac{p_k(\mathbf{x})}{p_{k+1}(\mathbf{x})} = \frac{p_0(\mathbf{x})}{p_m(\mathbf{x})}, \tag{4}$$

where $\boldsymbol{\theta}$ is the concatenation of all $\boldsymbol{\theta}_k$ vectors. Because of the telescoping product in (4), we refer to the method as Telescoping density-Ratio Estimation (TRE).

TRE has conceptual ties with a range of methods in optimisation, statistical physics and machine learning that leverage sequences of intermediate distributions, typically between a complex density $p$ and a simple tractable density $q$. Of particular note are the methods of Simulated Annealing [32], Bridge Sampling & Path Sampling [14] and Annealed Importance Sampling (AIS) [45]. Whilst none of these methods estimate density ratios, and thus serve fundamentally different purposes, they leverage similar ideas. In particular, AIS also computes a chain of density-ratios between artificially constructed intermediate distributions. It typically does this by first defining explicit expressions for the intermediate densities, and then trying to obtain samples via MCMC. In contrast, TRE *implicitly* defines the intermediate distributions via samples and then tries to learn the ratios. Additionally, in TRE we would like to evaluate the learned ratios in (4) at the *same* input $\mathbf{x}$ while AIS should only evaluate a ratio $r_k$ at 'local' samples from e.g. $p_k$.

## 3.1 Waymark creation

In this paper, we consider two simple, deterministic waymark creation mechanisms: *linear combinations* and *dimension-wise mixing*. We find these mechanisms yield good performance and are computationally cheap. However, we note that other mechanisms are possible, and are a promising topic for future work.

**Linear combinations.** Given a random pair $\mathbf{x}_0 \sim p_0$ and $\mathbf{x}_m \sim p_m$, define the $k^{\text{th}}$ waymark via

$$\mathbf{x}_k = \sqrt{1 - \alpha_k^2}\, \mathbf{x}_0 + \alpha_k \mathbf{x}_m, \qquad\qquad k = 0, \ldots, m \tag{5}$$

where the $\alpha_k$ form an increasing sequence from 0 to 1, which control the distance of $\mathbf{x}_k$ from $\mathbf{x}_0$. For all of our experiments (except, for illustration purposes, those depicted in Figure 1), each dimension of

$p_0$ and $p_m$ has the same variance[2] and the coefficients in (5) are chosen to preserve this variance, with the goal being to match basic properties of the waymarks and thereby make consecutive classification problems harder.

**Dimension-wise mixing.** An alternative way to 'mix' two vectors is to concatenate different subsets of their dimensions. Given a $d$-length vector $\mathbf{x}$, we can partition it into $m$ sub-vectors of length $d/m$, assuming $d$ is divisible by $m$. We denote this as $\mathbf{x} = (\mathbf{x}[1], \dots, \mathbf{x}[m])$, where each $\mathbf{x}[i]$ has length $d/m$. Using this notation, define the $k^{\text{th}}$ waymark via

$$\mathbf{x}_k = (\mathbf{x}_m[1], \ \dots, \ \mathbf{x}_m[k], \ \mathbf{x}_0[k+1], \ \dots, \ \mathbf{x}_0[m]) \qquad k = 0, \dots, m \qquad (6)$$

where, again, $\mathbf{x}_0 \sim p_0$ and $\mathbf{x}_m \sim p_m$ are randomly paired.

**Number and spacing.** Given these two waymark generation mechanisms, we still need to decide the *number* of waymarks, $m$, and, in the case of linear combinations, how the $\alpha_k$ are spaced in the unit interval. We treat these quantities as hyperparameters, and demonstrate in the experiments (Section 4) that tuning them is feasible with a limited search budget.

### 3.2 Bridge-building

Each bridge $r_k(\mathbf{x}; \boldsymbol{\theta}_k)$ in (4) can be learned via binary classification using a logistic loss function as described in Section 2. Solving this collection of classification tasks is therefore a multi-task learning (MTL) problem—see [55] for a review. Two key questions in MTL are how to *share parameters* and how to define a *joint objective function*.

**Parameter sharing.** We break the construction of the bridges $r_k(\mathbf{x}; \boldsymbol{\theta}_k)$ into two stages: a (mostly) shared body computing hidden vectors $f_k(\mathbf{x})$[3], followed by bridge-specific heads. The body $f_k$ is a deep neural network with shared parameters and pre-activation per-hidden-unit scales and biases for each bridge (see appendix for details). Similar parameter sharing schemes have been successfully used in the multi-task learning literature [7, 11]. The heads map the hidden vectors $f_k(\mathbf{x})$ to the scalar $\log r_k(\mathbf{x}; \boldsymbol{\theta}_k)$. We use either linear or quadratic mappings depending on the application; the precise parameterisation is stated in each experiment section.

**TRE loss function.** The TRE loss function is given by the average of the $m$ logistic losses

$$\mathcal{L}_{\text{TRE}}(\boldsymbol{\theta}) = \frac{1}{m} \sum_{k=0}^{m-1} \mathcal{L}_k(\boldsymbol{\theta}_k), \qquad (7)$$

$$\mathcal{L}_k(\boldsymbol{\theta}_k) = -\mathbb{E}_{\mathbf{x}_k \sim p_k} \log \left( \frac{r_k(\mathbf{x}_k; \boldsymbol{\theta}_k)}{1 + r_k(\mathbf{x}_k; \boldsymbol{\theta}_k)} \right) - \mathbb{E}_{\mathbf{x}_{k+1} \sim p_{k+1}} \log \left( \frac{1}{1 + r_k(\mathbf{x}_{k+1}; \boldsymbol{\theta}_k)} \right). \qquad (8)$$

This simple *unweighted* average works well empirically. More sophisticated multi-task weighting schemes exist [5], but preliminary experiments suggested they were not worth the extra complexity.

An important aspect of this loss function is that each ratio estimator $r_k$ sees different samples during training. In particular, $r_0$ sees samples close to the real data i.e. from $p_0$ and $p_1$, while the final ratio $r_{m-1}$ sees data from $p_{m-1}$ and $p_m$. This creates a potential mismatch between training and deployment, since after learning, we would like to evaluate all ratios at the *same* input $\mathbf{x}$. In our experiments, we do not find this mismatch to be a problem, suggesting that each ratio, despite seeing different inputs during training, is able to generalise to new test points. We speculate that this generalisation is encouraged by parameter sharing, which allows each ratio-estimator to be indirectly influenced by samples from *all* waymark distributions. Nevertheless, we think a deeper analysis of this issue of generalisation deserves further work.

### 3.3 TRE applied to mutual information estimation

The mutual information (MI) between two random variables $\mathbf{u}$ and $\mathbf{v}$ can be written as

$$I(\mathbf{u}, \mathbf{v}) = \mathbb{E}_{p(\mathbf{u}, \mathbf{v})} \left[ \log r(\mathbf{u}, \mathbf{v}) \right], \qquad\qquad r(\mathbf{u}, \mathbf{v}) = \frac{p(\mathbf{u}, \mathbf{v})}{p(\mathbf{u}) p(\mathbf{v})}. \qquad (9)$$

Given samples from the joint density $p(\mathbf{u}, \mathbf{v})$, one obtains samples from the product-of-marginals $p(\mathbf{u})p(\mathbf{v})$ by shuffling the $\mathbf{v}$ vectors across the dataset. This then enables standard density-ratio estimation to be performed.

For TRE, we require waymark samples. To generate these, we take a sample from the joint, $\mathbf{x}_0 = (\mathbf{u}, \mathbf{v}_0)$, and a sample from the product-of-marginals, $\mathbf{x}_m = (\mathbf{u}, \mathbf{v}_m)$, where $\mathbf{u}$ is held fixed and only $\mathbf{v}$ is altered. We then apply a waymark construction mechanism from Section 3.1 to generate $\mathbf{x}_k = (\mathbf{u}, \mathbf{v}_k)$, for $k = 0, \ldots, m$.

### 3.4 TRE applied to energy-based modelling

An energy-based model (EBM) is a flexible parametric family $\{\phi(\mathbf{x}; \boldsymbol{\theta})\}$ of non-negative functions, where each function is proportional to a probability-density. Given samples from a data distribution with density $p(\mathbf{x})$, the goal of energy-based modelling is to find a parameter $\boldsymbol{\theta}^*$ such that $\phi(\mathbf{x}; \boldsymbol{\theta}^*)$ is 'close' to $cp(\mathbf{x})$, for some positive constant $c$.

In this paper, we consider EBMs of the form $\phi(\mathbf{x}; \boldsymbol{\theta}) = r(\mathbf{x}; \boldsymbol{\theta})q(\mathbf{x})$, where $q$ is a known density (e.g. a Gaussian or normalising flow) that we can sample from, and $r$ is an unconstrained positive function. Given this parameterisation, the optimal $r$ simply equals the density-ratio $p(\mathbf{x})/q(\mathbf{x})$, and hence the problem of learning an EBM becomes the problem of estimating a density-ratio, which can be solved via TRE. We note that, since TRE actually estimates a product of ratios as stated in Equation 4, the final EBM will be a product-of-experts model [26] of the form $\phi(\mathbf{x}; \boldsymbol{\theta}) = \prod_{k=0}^{m-1} r_k(\mathbf{x}; \boldsymbol{\theta}_k)q(\mathbf{x})$.

The estimation of EBMs via density-ratio estimation has been studied in multiple prior works, including noise-contrastive estimation (NCE) [22], which has many appealing theoretical properties [22, 54, 65]. Following NCE, we will refer to the known density $q$ as the 'noise distribution'.

## 4 Experiments

We include two toy examples illustrating both the correctness of TRE and the fact that it can solve problems which verge on the intractable for standard density ratio estimation. We then demonstrate the utility of TRE on two high-dimensional complex tasks, providing clear evidence that it substantially improves on standard single-ratio baselines.

For experiments with continuous random variables, we use the linear combination waymark mechanisms in (5); otherwise, for discrete variables, we use dimension-wise mixing (6). For the linear combination mechanism, we collapse the $\alpha_k$ into a single spacing hyperparameter, and grid-search over this value, along with the number of waymarks. Details are in the appendix.

### 4.1 1d peaked ratio

The basic setup is stated in Figure 1a. For TRE, we use quadratic bridges of the form $\log r_k(x) = w_k x^2 + b_k$, where $b_k$ is set to its ground truth value (as derived in appendix), and $w_k$ is reparametrised as $\exp(\theta_k)$ to avoid unnecessary log-scales in Figure 1. The single ratio-estimation results use the same parameterisation (dropping the subscript $k$).

Figure 2 shows the full results. These sample efficiency curves clearly demonstrate that, across all sample sizes, TRE is significantly more accurate than single ratio estimation. In fact, TRE obtains a better solution with 100 samples than single-ratio estimation does with 100,000 samples: a three orders of magnitude improvement.

### 4.2 High-dimensional ratio with large MI

This toy problem has been widely used in the mutual information literature [2, 52]. Let $\mathbf{x} \in \mathbb{R}^{2d}$ be a Gaussian random variable, with block-diagonal covariance matrix, where each block is $2 \times 2$ with 1 on the diagonal and 0.8 on the off-diagonal. We then estimate the ratio between this Gaussian and a standard normal

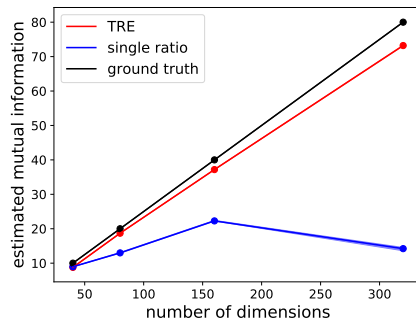

Figure 3: High-dimensional Gaussian results, showing estimated MI as a function of the dimensionality. Errors bars were computed over 5 random seeds, but are too small to see.

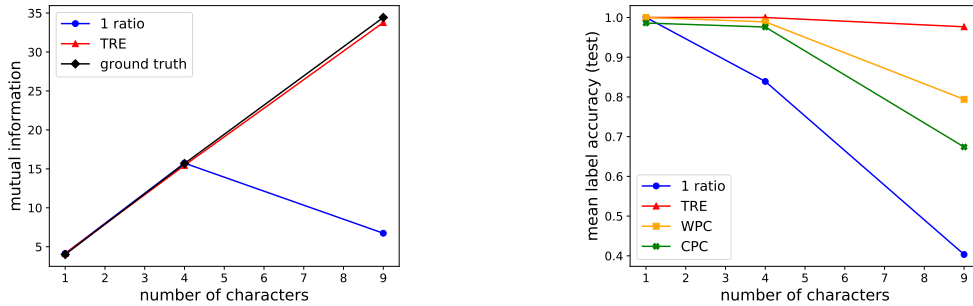

Figure 4: **Left**: mutual information results. TRE accurately estimates the ground-truth MI even for large values of $\sim 35$ nats. **Right**: representation learning results. All single density-ratio baselines (this includes CPC & WPC) degrade significantly in performance as we increase the number of characters from 4 to 9, dropping by 20-60% in accuracy. In contrast, TRE drops by only $\sim 3\%$.

distribution. This problem can be viewed as an MI estimation task or an energy-based modelling task—see the appendix for full details.

We apply TRE using quadratic bridges of the form: $\log r_k(\mathbf{x}) = \mathbf{x}^T \mathbf{W}_k \mathbf{x} + b_k$. The results in Figure 3 show that single ratio estimation becomes severely inaccurate for MI values greater than 20 nats. In contrast, TRE can accurately estimate MI values as large as 80 nats for 320 dimensional variables. To our knowledge, TRE is the first discriminative MI estimation method that can scale this gracefully.

### 4.3 MI estimation & representation learning on SpatialMultiOmniglot

We applied TRE to the SpatialMultiOmniglot problem taken from [49][4] where characters from Omniglot are spatially stacked in an $n \times n$ grid, where each grid position contains characters from a fixed alphabet. Following [49], the individual pixel values of the characters are not considered random variables; rather, we treat the grid as a collection of $n^2$ categorical random variables whose realisations are the characters from the respective alphabet. Pairs of grids, $(\mathbf{u}, \mathbf{v})$, are then formed such that corresponding grid-positions contain alphabetically consecutive characters. Given this setup, the ground truth MI can be calculated (see appendix).

Each bridge in TRE uses a separable architecture [52] given by $\log r_k(\mathbf{u}, \mathbf{v}) = g(\mathbf{u})^T \mathbf{W}_k f_k(\mathbf{v})$, where $g$ and $f_k$ are 14-layer convolutional ResNets [24] and $f_k$ uses the parameter-sharing scheme described in Section 3.2. We note that separable architectures are standard in the MI-based representation learning literature [52]. We construct waymarks using the dimension-wise mixing mechanism (6) with $m = n^2$ (i.e. one dimension is mixed at a time).

After learning, we adopt a standard linear evaluation protocol (see e.g. [48]), where we train *supervised* linear classifiers on top of the output layer $g(\mathbf{u})$ to predict the alphabetic position of each character in $\mathbf{u}$. We compare our results to those reported in [49]. Specifically, we report their baseline method—contrastive predictive coding (CPC) [48], a state-of-the-art representation learning method based on single density-ratio estimation—along with their variant, Wasserstein predictive coding (WPC).

Figure 4 shows the results. The left plot shows that only TRE can accurately estimate high MI values of $\sim 35$ nats[5]. The representation learning results (right) show that all single density-ratio baselines degrade significantly in performance as we increase the number of characters in a grid (and hence increase the MI). In contrast, TRE always obtains greater than 97% accuracy.

### 4.4 Energy-based modelling on MNIST

As explained in Section 3.4, TRE can be used estimate an energy-based model of the form $\phi(\mathbf{x}; \boldsymbol{\theta}) = \prod_{k=0}^{m-1} r_k(\mathbf{x}; \boldsymbol{\theta}_k) q(\mathbf{x})$, where $q$ is a pre-specified 'noise' distribution from which we can sample, and the product of ratios is given by TRE. In this section, we demonstrate that such an approach can

Table 1: Average negative log-likelihood in bits per dimension (bpd, smaller is better). Exact computation is intractable for EBMs, but we provide 3 estimates: Direct/RAISE/AIS. The 'Direct' estimate uses the NCE/TRE approximate normalising constant.

| Noise distribution | Noise | Single ratio (NCE) | | | TRE | | |
|---|---|---|---|---|---|---|---|
| | | Direct | RAISE | AIS | Direct | RAISE | AIS |
| Gaussian | 2.01 | 1.96 | 1.99 | 2.01 | 1.39 | 1.35 | 1.35 |
| Gaussian Copula | 1.40 | 1.33 | 1.48 | 1.45 | 1.24 | 1.23 | 1.22 |
| RQ-NSF | 1.12 | 1.09 | 1.10 | 1.10 | 1.09 | 1.09 | 1.09 |

|  | **Noise distribution** | **Single ratio (NCE)** | **TRE** |
|---|---|---|---|
| **Gaussian** | | | |
| **Copula** | | | |
| **RQ-NSF** | | | |

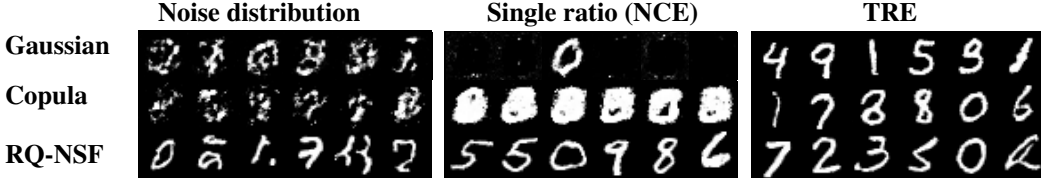

Figure 5: MNIST samples. Each row pertains to a particular noise distribution. The first block shows exact samples from that distribution. The second & third blocks show MCMC samples from an EBM learned with NCE & TRE, respectively.

scale to high-dimensional data, by learning energy-based models of the MNIST handwritten digit dataset [37]. We consider three choices of the noise distribution: a multivariate Gaussian, a Gaussian copula and a rational-quadratic neural spline flow (RQ-NSF) [12] with coupling layers [9, 31]. Each distribution is first fitted to the data via maximum likelihood estimation—see appendix for details.

Each of these noise distributions can be expressed as an invertible transformation of a standard normal distribution. That is, each random variable has the form $F(\mathbf{z})$, where $\mathbf{z} \sim \mathcal{N}(0, \mathbb{I})$. Since $F$ already encodes useful information about the data distribution, it makes sense to leverage this when constructing the waymarks in TRE. Specifically, we can generate linear combination waymarks via (5) in $\mathbf{z}$-space, and then map them back to $\mathbf{x}$-space, giving

$$\mathbf{x}_k = F(\sqrt{1 - \alpha_k^2}\, F^{-1}(\mathbf{x}_0) + \alpha_k F^{-1}(\mathbf{x}_m)). \tag{10}$$

For a Gaussian, $F$ is linear, and hence (10) is identical to the original waymark mechanism in (5).

We use the parameter sharing scheme from Section 3.2 together with quadratic heads. This gives $\log r_k(\mathbf{x}) = -f_k(\mathbf{x})^T \mathbf{W}_k f_k(\mathbf{x}) - f_k(\mathbf{x})^T \mathbf{b}_k - c_k$, where we set $f_k$ to be an 18-layer convolutional Resnet and constrain $\mathbf{W}_k$ to be positive definite. This constraint enforces an upper limit on the log-density of the EBM, which has been useful in other work [44, 46], and improves results here. We evaluate the learned EBMs quantitatively via estimated log-likelihood in Table 1 and qualitatively via random samples from the model in Figure 5. For both of these evaluations, we employ NUTS [28] to perform annealed MCMC sampling as explained in the appendix. This annealing procedure provides two estimators of the log-likelihood: the Annealed Importance Sampling (AIS) estimator [45] and the more conservative Reverse Annealed Importance Sampling Estimator (RAISE) [3].

The results in Table 1 and Figure 5 show that single ratio estimation performs poorly in high-dimensions for simple choices of the noise distribution, and only works well if we use a complex neural density-estimator (RQ-NSF). This illustrates the density-chasm problem explained in Section 2. In contrast, TRE yields improvements for all choices of the noise, as measured by the approximate log-likelihood and the visual fidelity of the samples. TRE's improvement over the Gaussian noise distribution is particularly large: the bits per dimension (bpd) is around 0.66 lower, corresponding to an improvement of roughly 360 nats. Moreover, the samples are significantly more coherent, and appear to be of higher fidelity than the RQ-NSF samples[6], despite the fact that TRE (with Gaussian noise) has a worse log-likelihood. This final point is not contradictory since log-likelihood and sample quality are known to be only loosely connected [61].

Finally, we analysed the sensitivity of our results to the construction of the waymarks and include the results in the appendix. Using TRE with a copula noise distribution as an illustrative case, we found that varying the number of waymarks between 5-30 caused only minor changes in the approximate log-likelihoods, no greater than $0.03$ bpd. We also found that if we omit the $\mathbf{z}$-space waymark mechanism in (10), and work in $\mathbf{x}$-space, then TRE's negative log-likelihood increases to $1.33$ bpd, as measured by RAISE. This is still significantly better than single-ratio estimation, but does show that the quality of the results depends on the exact waymark mechanism.

## 5   Conclusion

We introduced a new framework—Telescoping density-Ratio Estimation (TRE)—for learning density-ratios that, unlike existing discriminative methods, can accurately estimate ratios between extremely different densities in high-dimensions.

TRE admits many exciting directions for future work. Firstly, we would like a deeper theoretical understanding of why it is so much more sample-efficient than standard density-ratio estimation. The relationship between TRE and standard methods is structurally similar to the relationship between annealed importance sampling and standard importance sampling. Thus, exploring this connection further may be fruitful. Relatedly, we believe that TRE would benefit from further research on waymark mechanisms. We presented simple mechanisms that have clear utility for both discrete and continuous-valued data. However, we suspect more sophisticated choices may yield improvements, especially if one can leverage domain or task-specific assumptions to intelligently decompose the density-ratio problem. Lastly, whilst this paper has focused on the logistic loss, it would be interesting to more deeply investigate TRE with other discriminative loss functions.

## Broader Impact

As outlined in the introduction, density-ratio estimation is a foundational tool in machine learning with diverse applications. Our work, which improves density-ratio estimation, may therefore increase the scope and power of a wide spectrum of techniques used both in research and real-world settings. The broad utility of our contribution makes it challenging to concretely assess the societal impact of the work. However, we do discuss here two applications of density-ratio estimation with obvious potential for positive & negative impacts on society.

Generative Adversarial Networks [15] are a popular class of models which are often trained via density-ratio estimation and are able to generate photo-realistic image/video content. To the extent that TRE can enhance GAN training (a topic we do not treat in this paper), our work could conceivably lead to enhanced 'deepfakes', which can be maliciously used in fake-news or identity fraud.

More positively, density-ratio estimation is being used to correct for dataset bias, including the presence of skewed demographic factors like race and gender [18]. While we are excited about such applications, we emphasise that density-ratio based methods are not a panacea; it is entirely possible for the technique to introduce new biases when correcting for existing ones. Future work should continue to be mindful of such a possibility, and look for ways to address the issue if it arises.

## Acknowledgments and Disclosure of Funding

Benjamin Rhodes was supported in part by the EPSRC Centre for Doctoral Training in Data Science, funded by the UK Engineering and Physical Sciences Research Council (grant EP/L016427/1) and the University of Edinburgh. Kai was supported by Edinburgh Huawei Research Lab in the University of Edinburgh, funded by Huawei Technologies Co. Ltd.

## Footnotes

[1] 'nat' being a unit of information measured using the natural logarithm (base $e$)

[2]For MI estimation this always holds, for energy-based modelling this is enforceable via the choice of $p_m$.

[3]For simplicity, we suppress the parameters of $f_k$, and will do the same for $r_k$ in the experiments section.

[4]We mirror their experimental setup as accurately as possible, however we were unable to obtain their code.

[5][49] do not provide MI estimates for CPC & WPC, but [52] shows that they are bounded by log batch-size.

[6]We emphasise here that the quality of the RQ-NSF model depends on the exact architecture. A larger model may yield better samples. Thus, we do not claim that TRE generally yields superior results in any sense.

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
