[Supplementary Material]

# Telescoping Density-Ratio Estimation: Supplementary Material

**Benjamin Rhodes**
School of Informatics
University of Edinburgh
`ben.rhodes@ed.ac.uk`

**Kai Xu**
School of Informatics
University of Edinburgh
`kai.xu@ed.ac.uk`

**Michael U. Gutmann**
School of Informatics
University of Edinburgh
`michael.gutmann@ed.ac.uk`

## 1   ResNet architectures with parameter sharing

| |
|---|
| $5 \times 5$ conv with $3 \times 3$ strides, 32n |
| CondionalScaleShift |
| CondResBlock down, 32n |
| CondResBlock, 32n |
| CondResBlock down, 64n |
| CondResBlock, 64n |
| CondResBlock down, 64n |
| CondResBlock, 64n |
| GlobalSumPooling |
| Dense, 300n |

(a) SpatialMultiOmniglot architecture. The multiplier $n$ refers to the width/height of a datapoint, which is an $n \times n$ grid.

| |
|---|
| $3 \times 3$ conv, 64 |
| CondionalScaleShift |
| CondResBlock down, 64 |
| AttentionBlock |
| CondResBlock, 64 |
| CondResBlock down, 64 |
| CondResBlock, 64 |
| CondResBlock down, 128 |
| CondResBlock, 128 |
| CondResBlock down, 128 |
| CondResBlock, 128 |
| GlobalSumPooling |
| Dense, 128 |

(b) MNIST architecture.

(c) CondResBlock. Dashed boxes denote layers that are not always present. The $1 \times 1$ conv layer (and the associated CondScaleShift) is only used in blocks where the channel size is altered. The $2 \times 2$ pool layer is only used for 'down' blocks.

Figure 1: Convolutional ResNet architectures

In Figure 1, we give the exact architectures for the $f_k$ used in the two high-dimensional experiments on SpatialMultiOmniglot and MNIST. These $f_k$ output a hidden vector for the $k^{\text{th}}$ bridge, which is then mapped to the scalar value of the log-ratio, as stated in each experiment section. All convolution operations share their parameters across the bridges, and are thus independent of $k$.

The only difference between our conditional residual blocks (i.e. 'CondResBlocks') and a standard residual block is the use of 'ConditionalScaleShift' layers. These layers map a hidden vector $\mathbf{z}_k$ to a hidden vector of the same size, $\mathbf{z}'_k$, via

$$\mathbf{z}'_k = \mathbf{s}_k \odot \mathbf{z}_k + \mathbf{b}_k \tag{1}$$

where $\mathbf{s}_k$ and $\mathbf{b}_k$ are bridge-specific parameters and $\odot$ denotes element-wise multiplication. This operation could be thought of as class-conditional Batch Normalisation (BN) [3] without the normalisation. We did not investigate the use of BN, since many energy-based modelling papers (e.g. [4])

found it to harm performance. We did perform preliminary experiments with Instance Normalisation [20] in the context of energy-based modelling, finding it to be harmful to performance.

For the MNIST energy-based modelling experiments, we use average pooling operations since other work [19, 4] has found this to produce higher quality samples than max pooling. For the SpatialMultiOmniglot experiments, we grid-search over average pooling and max pooling. For both sets of experiments, we use LeakyRelu activations with a slope of 0.3.

The MNIST architecture includes an attention block [21] which has been used in GANs to model long-range dependencies in the input image. We found that this attention layer did not yield improvements in estimated log-likelihood, but we think it *may* yield slightly more globally coherent samples. We note that that another commonly used feature in recent GAN and EBM architectures is Spectral Normalisation (SN) [13]. Our preliminary experiments suggested that SN was not beneficial for performance. That said, all of our negative results should be taken with a grain of salt, given the preliminary nature of the experiments.

## 2   Waymark number and spacing

As stated in the main text, the number and (in the case of linear combinations) the spacing of the waymarks are treated as hyperparameters. Finding good values of these hyperparameters is made simpler by the following observations.

- If any of the TRE logistic losses saturate close to 0 during learning, then this indicates that the density-chasm problem has occured for that bridge, and we can terminate the run.
- As illustrated by our sensitivity analysis for MNIST (see Figure 5) it seems that, past a certain point, performance plateaus with the addition of extra waymarks. The fact that it plateaus, and does not decrease, is good news since it indicates that there is little risk of 'overshooting', and obtaining a bad model by having too many waymarks.

We now recall the linear combinations waymark mechanism, given by

$$\mathbf{x}_k = \sqrt{1 - \alpha_k^2}\,\mathbf{x}_0 + \alpha_k \mathbf{x}_m, \qquad\qquad k = 0, \ldots, m. \qquad (2)$$

where $m$ is the number of waymarks. We consider two ways of reducing the coefficients $\alpha_k$ to a function of a single spacing hyperparameter $p$ via

$$\alpha_k = (k/m)^p, \qquad\qquad k = 0, \ldots, m, \qquad (3)$$

$$\alpha_k = \left\{ \begin{array}{ll} (k/m)^p, & \text{for } k \leq m/2 \\ 1 - ((m-k)/m)^p, & \text{for } k \geq m/2 \end{array} \right\} \qquad k = 0, \ldots, m. \qquad (4)$$

Both mechanisms yield linearly spaced $\alpha_k$ when $p = 1$. For the first mechanism in (3), setting $p > 1$ means the gaps between waymarks *increase* with $k$ (and conversely decrease if $p < 1$). The spacing mechanism in (4) is a kind of symmetrised version of (3).

Table 1 shows the grid-searches we performed for all experiments. We note that these weren't always all performed in parallel. When using linear combinations, we typically set $p = 1$ initially and searched over values of $m$. If, for all values of $m$ tested, one of the TRE logistic losses saturated close to 0, then we would expand our search space and test different values of $p$.

Table 1: Waymark hyperparameters for each experiment. Curly braces {} denote grid-searches.

| experiment | mechanism | $m$ | spacing | $p$ |
|---|---|---|---|---|
| 1d peaked ratio | linear combo | 4 | Eq. 3 | $\{1, 2, \ldots, 7, 8\}$ |
| high dim, high MI | linear combo | $\frac{d}{40} \times \{1, 2, 3, 4\}$ | Eq. 3 | 1 |
| SpatialMultiOmniGlot | dim-wise mix | $d$ | N/A | N/A |
| MNIST (z-space) | linear combo | $\{5, 10, 15, 20, 25, 30\}$ | Eq. 3 | 1 |
| MNIST (x-space) | linear combo | $\{5, 10, 15, 20, 25, 30\}$ | Eq. { 3, 4 } | $\{1, 1.5, 2\}$ |

Note: $d$ refers to the dimensionality of the dataset, which is varied for certain experiments.

## 3   Minibatching

Recall that the TRE loss is a sum of logistic losses:

$$\mathcal{L}_{\text{TRE}}(\boldsymbol{\theta}) = \frac{1}{m} \sum_{k=0}^{m-1} \mathcal{L}_k(\boldsymbol{\theta}_k), \tag{5}$$

$$\mathcal{L}_k(\boldsymbol{\theta}_k) = -\mathbb{E}_{\mathbf{x}_k \sim p_k} \log \left( \frac{r_k(\mathbf{x}_k; \boldsymbol{\theta}_k)}{1 + r_k(\mathbf{x}_k; \boldsymbol{\theta}_k)} \right) - \mathbb{E}_{\mathbf{x}_{k+1} \sim p_{k+1}} \log \left( \frac{1}{1 + r_k(\mathbf{x}_{k+1}; \boldsymbol{\theta}_k)} \right). \tag{6}$$

When generating minibatch estimates of this loss, we can either sample from each $p_k$ independently, or we can *couple* the samples. By 'couple', we mean first drawing $B$ samples each from $p_0$ and $p_m$, randomly pairing members from each set, and then, for each pair, constructing all possible intermediate waymark samples to obtain a final minibatch of size $B \times M$. Coupling in this way means that the gradient of (5) w.r.t. $\boldsymbol{\theta}$ is estimated using shared sources of randomness, which can act as a form of variance reduction [14].

In all of our experiments, we use coupling when forming minibatches, since we found it to be useful in some preliminary investigations. However, coupling does have memory costs: the number of independent samples drawn from the data distribution, $B$, may need to be very small for the full minibatch, $B \times M$, to fit into memory. We speculate that as $B$ becomes sufficiently small, coupled minibatches will produce inferior results to non-coupled minibatches (which can use a greater number of independent real data samples). Empirical investigation of this claim is left to future work.

## 4   1d peaked ratio toy experiment

In this experiment we estimate the ratio $p_0/p_m$, where both densities are Gaussian, $p_0 = \mathcal{N}(0, \sigma_0^2)$ and $p_m = \mathcal{N}(0, \sigma_m^2)$, where $\sigma_0 = 10^{-6}$ and $\sigma_m = 1$. We generate waymarks using the linear combinations mechanism (2), which implies that each waymark distribution is Gaussian, since linear combinations of Gaussian random variables are also Gaussian. Specifically, the waymark distributions have the form

$$p_k(x) = \mathcal{N}(x; 0, \sigma_k^2), \qquad \text{where } \sigma_k = \left[ (1 - \alpha_k^2)\sigma_0^2 + \alpha_k^2 \sigma_m^2 \right]^{\frac{1}{2}}. \tag{7}$$

where the $\sigma_k$ form an increasing sequence between $\sigma_0$ and $\sigma_m$. The log-ratio between two waymark distributions is therefore given by

$$\log \frac{p_k(x)}{p_{k+1}(x)} = \log \left( \frac{\sigma_{k+1}}{\sigma_k} \right) + \left( \frac{1}{2\sigma_{k+1}^2} - \frac{1}{2\sigma_k^2} \right) x^2. \tag{8}$$

We parameterise the bridges in TRE as

$$\log r_k(x; \theta_k) = \log \left( \frac{\sigma_{k+1}}{\sigma_k} \right) - \exp(\theta_k)x^2, \tag{9}$$

where the quadratic coefficient $- \exp(\theta_k)$ is always negative. We note that this model is well-specified since it contains the ground-truth solution in (8).

The bridges can then be combined via summation to provide an estimate of the original log-ratio

$$\log \frac{p_0(x)}{p_m(x)} \approx \sum_{k=0}^{m-1} \log r_k(x; \theta_k) \tag{10}$$

$$= \log \left( \frac{\sigma_m}{\sigma_0} \right) - \sum_{k=0}^{m-1} \exp(\theta_k)x^2 \tag{11}$$

$$= \log \left( \frac{\sigma_m}{\sigma_0} \right) - \exp(\theta_{TRE})x^2 \tag{12}$$

Where $\theta_{TRE} = \log(\sum_{k=0}^{m-1} \exp(\theta_k))$. We observe that (12) has the same form as (9) if we were to set $m = 1$ in (9) (i.e. if we use a single bridge). Hence $\theta_{TRE}$ can be directly compared to the parameter value we would obtain if we used single density-ratio estimation. This is precisely the comparison we make in Figure 1a and Figure 2 of the main text.

## 4.1 The density chasm problem for non-logistic loss functions

In the main paper, we illustrated the density-chasm problem for the logistic loss using the 1d peaked ratio experiment. Here, we illustrate precisely the same phenomenon for the NWJ/MINE-f loss [16, 1] and a Least Squares (LSQ) loss used by [12]. The loss functions are given by

$$\mathcal{L}_{\text{NWJ}}(\boldsymbol{\theta}) = -\mathbb{E}_p \left[ \log r(\mathbf{x}; \boldsymbol{\theta}) \right] - 1 + \mathbb{E}_q \left[ r(\mathbf{x}; \boldsymbol{\theta}) \right] \tag{13}$$

$$\mathcal{L}_{\text{LSQ}}(\boldsymbol{\theta}) = \frac{1}{2}\mathbb{E}_p \left[ (\sigma(\log(r(\mathbf{x}; \boldsymbol{\theta}))) - 1)^2 \right] + \frac{1}{2}\mathbb{E}_q \left[ (\sigma(\log(r(\mathbf{x}; \boldsymbol{\theta}))))^2 \right], \tag{14}$$

where the $\sigma$ in (14) denotes the sigmoid function.

In Figures 2 & 3, we can see how single-density ratio estimation performs when using the NWJ and LSQ loss functions for 10,000 samples. the loss curves display the same 'saturation' effect seen for the logistic loss, where many settings of the parameter yield an almost identical value of the loss. Moreover, the minimiser of these saturated objectives is far from the 'true' minimiser (black dotted lines).

Figures 2 & 3 also show the performance of TRE when each bridge is estimated using the NWJ/LSQ losses. Each TRE loss has a quadratic bowl shape, where the finite-sample minimisers almost perfectly overlap with the true minimisers.

Finally, we plot sample efficiency curves for both the NWJ and LSQ losses, showing the results in Figure 4. We see that single density-ratio estimation with NWJ or LSQ performs poorly, with at best linear gains for exponential increases in sample size. In contrast, if we perform TRE using NWJ or LSQ losses, then we obtain significantly better performance with orders of magnitude fewer samples. These findings are essentially the same as those presented in the main paper for the logistic loss.

## 5 High-dimensional ratio with large MI toy experiment

In this experiment we estimate the ratio $p_0/p_m$, where both densities are Gaussian, $p_0 = \mathcal{N}(0, \Sigma)$ and $p_m = \mathcal{N}(0, \mathbb{I})$, where $\Sigma$ is a block-diagonal covariance matrix, where each block is $2 \times 2$ with 1 on the diagonal and $0.8$ on the off-diagonal. Since we know its analytic form, we can view $p_m$ as a noise distribution, and the ratio-estimation task as an energy-based modelling problem. Alternatively, we may view the problem as a mutual information estimation task, by taking the random variable $\mathbf{x} = (x_1, \ldots, x_d) \sim p_0$, and defining $\mathbf{u} = (x_1, x_3, \ldots, x_{d-1})$ and $\mathbf{v} = (x_2, x_4 \ldots x_d)$. By construction, we therefore have $p(\mathbf{u})p(\mathbf{v}) = \mathcal{N}(\mathbf{x}; 0, \mathbb{I}) = p_m(\mathbf{x})$.

We generate $100,000$ samples for each of the train/validation/test splits. We use a total batch size of $1024$, which includes all samples from the waymark trajectories. The bridges in TRE have the form $\log r_k(\mathbf{x}) = \mathbf{x}^T \mathbf{W}_k \mathbf{x} + b_k$, where we enforce that the diagonal entries of $\mathbf{W}_k$ are positive and that the matrix is symmetric. We use the Adam optimiser [8] with an initial learning rate of $0.0001$ for TRE, and $0.0005$ for single ratio estimation. We use the default Tensorflow settings for $\beta_1, \beta_2$ and $\epsilon$. We train the models for $40,000$ iterations, which takes at most 1 hour.

## 6 MI estimation & representation learning on SpatialMultiOmniglot

We here describe how we created the SpatialMultiOmniglot dataset and give the derivation for the ground truth mutual information values presented in the main paper[1]. We will share the dataset, along with code for the paper, upon publication. We also state the hyperparameter settings used in our experiments.

### 6.1 Dataset construction

We take the Tensorflow version of the Omniglot dataset (`https://www.tensorflow.org/datasets/catalog/omniglot`) and resize it to $28 \times 28$ using the `tf.image.resize` function. We arrange the data into alphabets $\{A_i\}_{i=1}^l$, where each alphabet contains $n_i$ characters. The alphabets are sorted by size, so that $n_1 > n_2 > \ldots > n_l$. Each character in a alphabet has 20 different

## NWJ loss

$$\frac{p}{q} = \frac{p}{p_1} \times \frac{p_1}{p_2} \times \frac{p_2}{p_3} \times \frac{p_3}{q}$$

Figure 2: Replica of Figure 1 from the main text, except that we use the NWJ/MINE-f loss [16, 1] for both the single ratio estimator & for each ratio in TRE.

## Least-square loss

$$\frac{p}{q} = \frac{p}{p_1} \times \frac{p_1}{p_2} \times \frac{p_2}{p_3} \times \frac{p_3}{q}$$

Figure 3: Replica of Figure 1 from the main text, except that we use the least-square loss from the GAN literature [12] for both the single ratio estimator & for each ratio in TRE.

Figure 4: Sample efficiency curves for the 1d peaked ratio experiment, using different loss functions.

*versions* (e.g. there are 20 different images depicting the letter 'w'). Hence, we can express each alphabet as a set $A_i = \{\{a_{j,k}^i\}_{k=1}^{20}\}_{j=1}^{n_i}$, where $a_{j,k}^i$ refers to the $k^{\text{th}}$ version of the $j^{\text{th}}$ character of the $i^{\text{th}}$ alphabet.

In order to construct the $d$-dimensional version of the SpatialMultiOmniGlot dataset, we restrict ourselves to the $d$ largest alphabets $\{A_i\}_{i=1}^d$. We then sample a vector of categorical random variables

$$\mathbf{j} = (j_1, \ldots, j_d) \sim \text{Cat}(n_1) \times \ldots \text{Cat}(n_d) \tag{15}$$

where the $i^{\text{th}}$ categorical distribution is uniform over the set $\{1, \ldots, n_i\}$ and is independent from the other categorical distributions. The vector $\mathbf{j}$ should be thought of as an index vector that specifies a particular character from each of the $d$ alphabets.

We then sample two i.i.d random variables $\mathbf{k}$ and $\mathbf{k}'$, via

$$\mathbf{k} = (k_1, \ldots, k_d) \sim \prod_{i=1}^d \text{Cat}(20) \qquad \mathbf{k}' = (k_1', \ldots, k_d') \sim \prod_{i=1}^d \text{Cat}(20) \tag{16}$$

where, again, each Categorical distribution is independent from the rest. These vectors should be thought of as index vectors that specify a particular version of a character.

Now, we define a datapoint as a tuple $\mathbf{x} = (\mathbf{u}, \mathbf{v})$, where

$$\mathbf{u} = (a_{j_1, k_1}^1, \ldots, a_{j_d, k_d}^d) \qquad \mathbf{v} = (a_{j_1+1, k_1'}^1, \ldots a_{j_d+1, k_d'}^d). \tag{17}$$

In words, we construct $\mathbf{u}$ and $\mathbf{v}$ such that $u_i$ and $v_i$ are consecutive characters within their alphabet (whilst the precise *versions* of the characters are randomised). Finally, we arrange $\mathbf{u}$ and $\mathbf{v}$ into a grid using raster ordering. This is possible since we assume $d$ to be a square number.

Importantly, we emphasise that $\mathbf{u}, \mathbf{v} \in \prod_{i=1}^d A_i$ are discrete random variables defined over a set of template images. They are not defined over a space of pixel values, as is usually the case in image-modelling.

## 6.2  Derivation of ground truth MI

By construction, we have that $\mathbf{u}$ and $\mathbf{v}$ are conditionally independent given $\mathbf{j}$. This means

$$p(\mathbf{u}|\mathbf{v}, \mathbf{j}) = p(\mathbf{u}|\mathbf{j}). \tag{18}$$

Furthermore, will assume that, for all $\mathbf{u}$ there exists a unique $\mathbf{j_u}$ such that

$$p(\mathbf{j_u}|\mathbf{u}) = 1. \tag{19}$$

Similarly, for any $\mathbf{v}$, there exists a unique $\mathbf{j_v}$ satisfying the same condition. In words, this simply means that, given a grid of Omniglot images, we assume there is no ambiguity about which characters are present. Using Bayes' rule, and the fact that for a given $\mathbf{j}$, $\mathbf{u}$ is uniquely determined by $\mathbf{k}$, one can then deduce that

$$p(\mathbf{u}|\,\mathbf{j}) = \left\{ \begin{array}{ll} 0, & \text{for } \mathbf{j} \neq \mathbf{j_u} \\ 20^{-d}, & \text{for } \mathbf{j} = \mathbf{j_u} \end{array} \right\}. \tag{20}$$

and similarly for $\mathbf{v}$.

We now proceed to derive an analytical formula for the ground truth mutual information between $\mathbf{u}$ and $\mathbf{v}$. We show that the mutual information is equal to the sum of the log alphabet sizes $\mathcal{I}(\mathbf{u}, \mathbf{v}) = \sum_{i=1}^{d} \log n_i$.

This holds because

$$\mathcal{I}(\mathbf{u}, \mathbf{v}) = \mathbb{E}_{p(\mathbf{u},\mathbf{v})} \log \frac{p(\mathbf{u}, \mathbf{v})}{p(\mathbf{u})p(\mathbf{v})} \tag{21}$$

$$= \mathbb{E}_{p(\mathbf{u},\mathbf{v})} \log \frac{p(\mathbf{u}|\,\mathbf{v})}{p(\mathbf{u})} \tag{22}$$

$$= \mathbb{E}_{p(\mathbf{u},\mathbf{v})} \log \frac{\sum_{\mathbf{j}} p(\mathbf{u},\mathbf{j}|\,\mathbf{v})}{\sum_{\mathbf{j}} p(\mathbf{u},\mathbf{j})} \tag{23}$$

$$= \mathbb{E}_{p(\mathbf{u},\mathbf{v})} \log \frac{\sum_{\mathbf{j}} p(\mathbf{u}|\,\mathbf{j})p(\mathbf{j}|\,\mathbf{v})}{\sum_{\mathbf{j}} p(\mathbf{u}|\,\mathbf{j})p(\mathbf{j})} \qquad \text{by (18)} \tag{24}$$

$$= \mathbb{E}_{p(\mathbf{u},\mathbf{v})} \log \frac{p(\mathbf{u}|\,\mathbf{j_v})}{\sum_{\mathbf{j}} p(\mathbf{u}|\,\mathbf{j})p(\mathbf{j})} \qquad \text{by (19)} \tag{25}$$

$$= \mathbb{E}_{p(\mathbf{u},\mathbf{v})} \log \frac{p(\mathbf{u}|\,\mathbf{j_v})}{p(\mathbf{u}|\,\mathbf{j_u})p(\mathbf{j_u})} \qquad \text{by (20)} \tag{26}$$

$$= \mathbb{E}_{p(\mathbf{u},\mathbf{v})} \log \frac{1}{p(\mathbf{j_u})} \qquad \text{since } \mathbf{j_v} = \mathbf{j_u} \text{ if } p(\mathbf{u}, \mathbf{v}) > 0 \tag{27}$$

$$= \mathbb{E}_{p(\mathbf{u},\mathbf{v})} \log \Big( \prod_{i=1}^{d} n_i \Big) \qquad \text{since } p(\mathbf{j}) \text{ is uniform} \tag{28}$$

$$= \sum_{i=1}^{d} \log n_i \tag{29}$$

### 6.3 Experimental settings

We generate 3 versions of the SpatialMultiOmniglot dataset for $d = 1, 4, 9$. For each version, we sample $50,000$ training points and $10,000$ validation and test points. As stated in the main text, we use a separable architecture given by

$$\log r_k(\mathbf{u}, \mathbf{v}) = g(\mathbf{u})^T \mathbf{W}_k f_k(\mathbf{v}), \tag{30}$$

where $f_k$ is a convolutional ResNet whose architecture is given in Figure 1a. The function $g$ is also a convolutional ResNet with almost the same architecture, except that none of its parameters are bridge-specific, and hence the 'ConditionalScaleShift' layers simply become 'ScaleShift' layers, with no dependence on $k$.

To construct a mini-batch, we first sample a batch from the joint distribution $p(\mathbf{u}, \mathbf{v})$. We then obtain samples from $p(\mathbf{u})p(\mathbf{v})$ by sampling a *second* batch from the joint distribution (which could overlap with the first batch), and shuffling the $\mathbf{v}$ vectors across this second batch. Finally, we construct waymark trajectories as described in the main text. For all experiments, the 'total' batch size is $\sim 512$, which includes all samples from the waymark trajectories. Thus, as the number of waymarks increases, the number of trajectories in a batch decreases.

We use the Adam optimiser [8] with an initial learning rate of $10^{-4}$ with default Tensorflow settings for $\beta_1, \beta_2$ and $\epsilon$. We gradually decrease the learning rate over the course of training with cosine

annealing [11]. All models are trained using a single NVIDIA Tesla P100 GPU card for $200,000$ iterations, which takes at most a day.

We grid-searched over the type of pooling (max vs. average) and the size of the final dense layer ($150n, 300n$ and $450n$, where $d = n^2$). Interestingly, average pooling was less prone to overfitting and often yielded better final performance, however it was often 'slow to get started', with the TRE losses hardly making any progress during the first quarter of training.

For the representation learning evaluations, we first obtained the hidden representations $g(\mathbf{u})$ for the entire dataset. We then trained a collection of *independent* supervised linear classifiers on top of these representations, in order to predict the alphabetic position of each character in $\mathbf{u}$. We used the L-BFGS optimiser to fit these classifiers via the `tfp.optimizer.lbfgs_minimize` function, setting the maximum iteration number to $10,000$.

# 7 Energy-based modelling on MNIST

We here discuss the parameterisation of the noise distributions used in the experiments, the exact method for sampling from the learned EBMs, and the experimental settings used for TRE.

For all noise distributions and TRE models, we use the Adam optimiser [8] with an initial learning rate of $10^{-4}$ with default Tensorflow settings for $\beta_1, \beta_2$ and $\epsilon$. We gradually decrease the learning rate over the course of training with cosine annealing [11]. All models are trained using a single NVIDIA Tesla P100 GPU card.

## 7.1 Noise distributions

As stated in the main text, we consider three noise distributions: a multivariate Gaussian, a Gaussian copula and a rational-quadratic neural spline flow (RQ-NSF), all of which are pre-trained via maximum likelihood estimation.

The full-covariance multivariate Gaussian is by far the simplest, and can be fitted in around a minute via `np.cov`. The Gaussian copula is slightly more complicated. Its density can be written as $p(\mathbf{x}) = \mathcal{N}([s_1(x_1), \ldots, s_d(x_d)]; \mu, \Sigma) \prod_{i=1}^{d} |s'_i(x_i)|$. The $s_i$ are given by the composition of the inverse CDF of a standard normal and the CDF of the univariate $x_i$. It is possible to exploit this to learn the $s_i$—as well as $\mu$ and $\Sigma$—however, we found it slightly simpler to directly parametrise the $s_i$ via flexible rational-quadratic spline functions [5] of which there are official implementations in Tensorflow and Pytorch and to jointly learn all parameters via maximum likelihood. We follow the basic hyperparameter recommendations in [5]. The hyperparameters that required tuning were the number of bins (we use 128) and the interval widths (which we set to 3 times the standard deviation of the data). For optimisation, we used a batch size of 512 and trained for $40,000$ iterations.

Finally, we turn to the RQ-NSF model [5]. We largely adopt the architectural choices of [5], and so for a more detailed explanation, we refer the reader to their work. We use a multi-scale convolutional architecture comprised of 2 levels, where each level contains 8 'steps'. A step consists of an actnorm layer, an invertible $1 \times 1$ convolution, and a rational-quadratic coupling transform. The coupling transforms are parameterised by a block of convolution operations following [9], which use 64 feature maps. The spline functions use 8 bins and the interval width is set to [-3, 3]. We do not 'factor out' half of the variables at the end of each level, but do perform 'squeeze' operation and an additional $1 \times 1$ convolution. For optimisation, we set the batch size to 256, the dropout rate to 0.1, and train for $200,000$ iterations, which takes under a day.

## 7.2 Annealed MCMC Sampling

We here describe how we leverage the specific products-of-experts structure of the TRE model to perform annealed MCMC sampling. Firstly, we initialise a set of MCMC chains with i.i.d samples from the noise distribution $p_m$. We could then run an MCMC sampler with the full TRE model as the target distribution. However, we instead use an annealing procedure, whereby we iteratively sample from a sequence of distributions that interpolate between $p_m$ and $p_0$. Such distributions can

be obtained by multiplying $p_m$ with an increasing number of bridges

$$p_k(\mathbf{x}) = p_m(\mathbf{x}) \prod_{i=k}^{m-1} r_k(\mathbf{x}), \qquad\qquad k = m-1, \ldots 0. \qquad (31)$$

To obtain an even smoother interpolation, we further define exponentially-averaged intermediate distributions $p_{k,t}(\mathbf{x}) = p_k(\mathbf{x})^{\beta_t} p_{k+1}(\mathbf{x})^{1-\beta_t}$, where $\{\beta_t\}$ is a decreasing sequence of numbers ranging from 1 to 0.

In addition to obtaining samples, we can simultaneously use this annealing procedure for estimating the log-likelihood of the model via annealed importance sampling (AIS) [15]. We may also run the annealing procedure 'in reverse', initialising a chain at a datapoint and iteratively removing bridges until the target distribution of the MCMC sampler is the noise distribution. Using this reverse sampling procedure, we can obtain a second, more conservative, estimate of the log-likelihood via the reverse annealed importance sampling estimator (RAISE) [2].

Whilst in principle any MCMC sampler could be used, the efficiency of different samplers can vary greatly. We choose to use the gradient-based No-U-turn sampler (NUTS) [7], which is a highly efficient method for many applications. We use the official Tensorflow implementation along with most of the default hyperparameter settings. We set the target acceptance rate to 0.6, and use a max tree depth of 6 during the annealed sampling. We also continue to run the sampler after the annealing phase is finished, using a max tree depth of 10. We use a total of 1000 intermediate distributions with 100 parallel chains.

Finally, recall from the main text that each noise distribution in our experiments can be expressed as invertible transformation $F$ of a standard normal distribution. We use this $F$ to further enhance the efficiency of the NUTS sampler, by performing the sampling in the $\mathbf{z}$-space, and then mapping the final results back to $\mathbf{x}$-space. Working in $\mathbf{z}$-space, by the rules of transformations of random variables, the intermediate distributions of (31) become

$$p_k(\mathbf{z}) = \mathcal{N}(\mathbf{z}; 0, \mathbb{I}) \prod_{i=k}^{m-1} r_k(F(\mathbf{z})). \qquad (32)$$

AIS and RAISE can still be applied, just as before, to obtain an estimate of the log-likelihood in $\mathbf{z}$-space. The change of variables formula for probability density functions can then be applied to obtain estimated log-likelihoods for the original TRE model in $\mathbf{x}$-space. We note that when the noise distribution is a normalising flow, prior work has demonstrated that $\mathbf{z}$-space MCMC sampling can be significantly more effective than working in the original data space [6].

### 7.3 Experimental settings

We use the standard version of the MNIST dataset [10], with $50,000$ training points, and $10,000$ validation and test points. We follow the same preprocessing steps as [18], 'dequantizing' the dataset with uniform noise, re-scaling to the unit interval, and then mapping to the real line via a logit transformation.

The architecture for the TRE bridges is given in Figure 1b. The waymark mechanism and associated grid-search is given in Table 1. A consistent observation across all our MNIST experiments was that the first ratio-estimator between the data distribution $p_0$ and a slightly perturbed data distribution $p_1$ was extremely prone to overfitting. We found that the only way to mitigate this problem was to simply drop the ratio by setting the $\alpha_0$ in (2) to a very small value (0.01) rather than exactly 0. Equivalently, this can be viewed as applying standard TRE to a very slightly perturbed data distribution. We note that this perturbation is small enough that is barely visible to the human eye when comparing samples. We conjecture that this problem may stem from the fact that the original MNIST dataset is actually discrete not continuous and the 'dequantizing' perturbation used to make the data continuous is perhaps not sufficient.

To form mini-batches, we sample 25 datapoints each from $p_0$ and $p_m$, and then generate waymark trajectories as described in the main text. Thus, the total batch size is $25 \times (m+1)$. We use the optimisation settings described at the beginning of this section, training for $200,000$ iterations, which takes about a day.

## 7.4 Additional results

In Figure 5, we present a sensitivity analysis showing how the quality of the learned EBM varies as we alter the number of waymarks, as well as the space in which the waymarks are generated. We found that working in $\mathbf{x}$-space yielded lower performance compared to working in $\mathbf{z}$-space, as measured by the most conservative estimator, RAISE. In particular, we found that the $\mathbf{x}$-space mechanism required more waymarks (around 15) to avoid any of the logistic losses saturating close to 0, and it was significantly harder to tune the spacing of the waymarks as indicated by Table 1.

Finally, for the models whose results were given in the main paper, we display extended image samples in Figure 6. Note that these samples are ordered by log-density (lowest density in top left corner, highest in bottom right).

Figure 5: Waymark sensitivity analysis for TRE with copula noise distribution. Both plots show the estimated bits per dimension (bpd) as a function of waymark number. On the left we apply the linear combination waymark mechanism in $\mathbf{z}$-space, whilst on the right we apply it in $\mathbf{x}$-space. As described in Section 2, we terminate runs where any of the TRE losses saturate close to 0, which is exactly what happened when using the $\mathbf{x}$-space mechanism for 5 and 10 waymarks.

(a) Samples from TRE model with Gaussian noise distribution, ordered by log-density.

(b) Samples from TRE model with copula noise distribution, ordered by log-density.

(c) Samples from TRE model with RQ-NSF noise distribution, ordered by log-density.

Figure 6: Extended MNIST samples

## Footnotes

[1]The original work from which we borrow this experiment [17] did not not provide a detailed explanation or code.