[Reviews · NeurIPS 2020]

Review 1

Summary and Contributions: This paper introduces a clever idea for estimating density ratios via discriminative methods in a regime where they counter-intuitively fail: when distributions are easy to distinguish because they are far apart. Important problems like KL divergence and mutual information estimation require modeling density ratios, and while discriminative approaches have been successful in these cases, they are widely known to have the problem that is addressed by this paper. The paper suggests bridging the gap between distributions with a sequence of intermediate distributions, and explores the best way to construct the bridge and to parametrize the family of discriminators that is now required. The paper demonstrates reasonable results on two relevant downstream tasks: mutual information estimation and density modeling.

Strengths: - The basic idea for bridging the gap is elegant and seems to differ in implementation from the standard "bridging" techniques in annealed importance sampling, though more discussion of that would be appreciated. - I have encountered versions of the "density chasm" problem, as the authors formulate it, many times in information estimation, so I find new solutions to be useful. (Approaches based on transporting densities like Wasserstein, are not as easily combined with other methods as this approach, in my experience). - Mutual information estimation and density modeling were sensible choices for evaluation. While the MI estimation tasks look somewhat artificial, I appreciate the difficulty of finding non-trivial examples where the ground truth is known. It is good to see the approach shines in this particular case.

Weaknesses: The main issue for me was the presentation and empirical results for the energy-based modeling. - MNIST is rather easy to model, so while the results were instructive as demonstrations, they don't really hint whether the approach can work in more complicated domains. - My main confusion was with the framing of the approach as energy-based modeling. In the first few mentions, it wasn't obvious why the approach was relevant for this problem. Then the introduction of EBMs is very brief and gives no reminder of the noise contrastive estimation setup. I had to refresh myself on NCE before this section made sense. I think it would be better to say, throughout, that this idea can be used for NCE, and then remind people that NCE is a way to learn unnormalized/energy-based models. NCE is somewhat special compared to the EBMs that first came to mind, since you have to specify the noise distribution. Since I hadn't looked at NCE lately, the sudden introduction of noise distributions without context was hard to understand.

Correctness: Yes, the derivation is straightforward.

Clarity: Yes, except for the introduction of energy-based models noted above, which I believe is easily fixed.

Relation to Prior Work: Yes, the related work is thorough. More discussion of the relationships (AIS and NCE) would be appreciated though.

Reproducibility: Yes

Additional Feedback: - I was making a note about the relationship with annealed importance sampling when I got to the conclusion and saw that you suggest this for future work. The parallels are interesting, I hope this direction is taken up. There's been some recent work on "thermodynamic variational inference" that involves a similar type bridging that you may also consider. - I didn't closely study the Stratos & MacAllester paper "Formal Limitations...", but they seemed to be making general claims about the impossibility of MI estimation with large MIs that Fig 4 suggests you are not subject to. Is it easy to say if the limitations in that paper apply to your method and, if not, why not? - A very recent paper that you couldn't have included but may be relevant: Gao et. al,  Flow Contrastive Estimation of Energy-Based Models, CVPR 2020. Edit: I read the rebuttal.


Review 2

Summary and Contributions: The paper targets the problem of density-ratio estimation via discriminative classification for cases of large differences in the densities. In these cases, classification is easy enough s.t. discriminative classifiers do not have to represent the densities accurately, thus leading to high errors when used for density ratio estimation. The paper proposes a telescoping mechanism where one density is stepwise transformed towards the other. Classification between these very similar intermediate densities is much harder and thus the discriminative classifier is forced towards more accurate density estimates.

Strengths: The idea of the paper seems both straightforward and effective, which is interesting. The experiments show estimates that are close to the ground-truth even for `problematic`data. Fig. 4 (right) shows strong benefits of the introduced approach compared to a very recent NeurIPS paper. Interestingly the required number of samples decreases significantly, since it is no longer important to acquire samples from low density regions (which takes an exponentially increasing number of samples when assuming an exponential model as in Fig.2).

Weaknesses: Many design choices appear quite ad hoc. With little analytical guarantees and only few experiments (most of them involving variants of Gaussian noise) it is hard to assess how generalizable the results are. No comparisons to models other than "single ratio" estimations are given. It raises the question, why this problem was never addressed before, or if it was addressed before why there is no comparison.

Correctness: I am not an expert in the field of density ratio estimation. I found it hard to assess if the claimed severity of the proposed "density-chasm" is valid and the proposed solution is therefore relevant: On one hand it is claimed that the "density-chasm proplem" easily makes calculations in high dimensions "virtually intractable with existing techniques". On the other hand, this does not seem to hinder state-of-the-art results in many applications. (E.g. many papers are later cited that learn EBMs in high dimensions - or in other words: in which cases is it really important if the ratio is estimated as 10^12 when it actually is 10^10?) [51] is cited to denote large MI values as an "outstanding" problem in MI estimation. The cited paper itself talks about "challenging". [51] is also cited for the experimental setup in 4.2, but no experimental comparison to [51] or further discussion of the approach is given. An overview over state-of-the-art methods which also address this "severe limitation" and how the proposed model relates to those is not given.

Clarity: There are some mildly confusing points in the figures and experimental descriptions; some parameters are not properly explained (e.g. the 1-dimensional \theta). (see comments) The severity of the addressed problem and the reason why there is no other algorithm that addesses it is not made clear enough. Otherwise the paper is clearly written.

Relation to Prior Work: It is not clear which other approaches exist that address the proposed "density-chasm problem" and how those relate to this work. TRE is only compared to single ratio models, that do not address this problem. If there are no other models that address this problem, it should be explained why that is the case.

Reproducibility: Yes

Additional Feedback: Comments are numbered w.r.t. the sections in the submitted work that they refer to. 2. a) The main idea of the framework is easy to understand and seems like a good idea. The critical question for me is how well the error accumulation via the telescoping product behaves under which circumstances. I would imagine that the total relative error of the telescope product should scale with the sum of the individual relative errors of each term. Using more waymarks should then decrease the relative error per term, but increase the number of terms and with that the total error. This total error would then also highly depend on the quality of the chosen waymarks/bridges. The paper however gives little information on the motivation for the concrete choices (which seem very dependent on the experiment and model). b) What did you use as free parameter \theta (line 88 and Fig.1)? c) The axes of Fig. 1(a, left) are confusing. Maybe use axis breaks to denote where you switch from logarithmic to linear scale to show the 0. d) The axes scaling/cut makes p(x) invisible. This should be either corrected or noted in the caption. 3.1 Waymark creation: "For our applications, each dimension of p_0 and p_m has the same variance" Can you elaborate on this? In Fig. 1(a) p(x) has sigma=1e-6 and q(x) has sigma=1 (maybe I misunderstand what variance you refer to). Could you provide a derivation of Eq. (5) from equal variances (e.g. in the appendix)? It seems a bit odd to use such an equation (and call it linear) instead of using (1-a)x_0 + a*x_m. Since the waymarks are proposed ad hoc, it is unclear what choice of waymarks is suitable for which kind of distributions. It would be helpful to note already here that dimension-wise mixing is for discrete cases and the linear combination for continuous cases (if I understood correctly). That would make the motivation and use cases for the two clearer. 4. Experiments: It would be helpful to refer to App. 2 for the ad hocly introduced "extra spacing parameter". Please also refer to the other appendices at the appropriate places to mark where additional information is provided. 4.1 The "quadratic bridges" seem like a reformulation of scaled Gaussians with zero mean: log(r(x)) = b_k + w_k*x^2 -> r(x) = A*exp(x^2/sigma^2). Is this association correct (and similar for the bridges with covariance matrix)? Is this the reason why these kind of bridges work well for the Gaussian toy example in 4.2 and the Gaussian noise experiments in 4.4? 4.2 You write: "To our knowledge, TRE is the first discriminative MI estimation method that can scale this gracefully.", but you do not show any comparisons to competing methods other than single ratio, despite stating that "this toy problem has been widely used". Comparisons should then be available and should be shown. 4.3 SpatialOmniglot a) It is hard to understand from the description what exactly you are doing (I had to look into [48] to understand the setup). Maybe a visualization of the data set would help (there is lots of space in Figure 4). It is also confusing that you use the word "stacked", since there is also a StackedMultiOmniglot, which you don't seem to use. b) What motivated this choice of network? c) The results in Fig.4 (right) look very convincing. How far can TRE go until the performance drops? Is it feasible to show experiments with e.g. 25 characters? 4.4 "the quality of the results depends on the exact waymark mechanism" The introduced mechanism all appear very ad hoc. How did you decide between the one or another? Is this decision purely empirically motivated or are there analytical reasons that support these choices? Note: I am willing to go up with my score if my questions, especially regarding the relevance/severity of the addressed problem and relations to competing methods, can be answered satisfactory. Answer to Rebuttal: I have read the author feedback and found the answers satisfactory. Since I am now more convinced about the relevance of the adressed problem, and since I still find the solution very appealing, I raised my score for the submission and now vote for a clear accept.


Review 3

Summary and Contributions: This paper proposes a better classifier-based estimator for density ratios. The idea is based on annealing. Instead of classify between two distributions, the authors propose to classify between multiple bridge distributions and estimate the density ratio by combining all the intermediate estimates. This approach is particularly suitable to solve the density-chasm problem, where two distributions are very far away from each other and the classification problem becomes too easy. Experiments demonstrate great potential of this method in mutual information estimation and learning energy-based models.

Strengths: 1. Density ratio estimation is very important in various aspects of machine learning, including representation learning, mutual information estimation, and learning energy-based generative models. It is well known that classifier-based methods for density ratio estimation have bad performance whenever two distributions are too far away from each other, and this has led to many problems, for example the difficulty of choosing the proposal distributions in noise-contrastive estimation, and the excessive variance in classifier-based mutual information estimation. A better density ratio estimator can have broad impact in many applications and is very interesting to the general NeurIPS community. 2. The proposed method is applicable to discrete random variables, which is not always true for competing methods. 3. Experiments convincingly demonstrate the advantages of telescoping over the baseline.

Weaknesses: 1. Both waymark creation and bridge-building can be tricky to tune. Finding good intermediate samples are hard and have to rely heavily on experimentation and experience, and special tricks like coupling sources of randomness is needed for variance reduction. Unfortunately the experimental results in Section 4.4 demonstrates that choosing right waymarks is indeed critical for the performance (x space vs. latent space). Aside from waymark creation, it is also tricky to find a correct architecture of the estimator to share information across different bridge distributions. 2. It is useful to have theoretical analysis on why telescoping significantly improves the performance of single-ratio density estimation. Since experiments in this paper are on relatively low data space, it is also valuable to know how the telescoping method scales to higher dimensions. Would higher dimension problems require considerably more intermediate distributions? If so, how bad is it?

Correctness: No obvious mistakes as far as I know.

Clarity: It is a very well-written paper and enjoyable to read.

Relation to Prior Work: Yes

Reproducibility: Yes

Additional Feedback: ---------post rebuttal--------- Thanks the authors for the rebuttal and there is no need of changing my current rating.


Review 4

Summary and Contributions: Density ratio estimation is hard when the involved densities diverge a lot. The surrogate objective (discriminator) can easily distinguish samples from them and no signal is received. The paper addresses this problem by introducing intermediate distributions in the tradition of adaptive importance sampling or bridge sampling. For these distributions, density ratio estimators can be trained and the product of density ratio estimates give the overall density ratio.

Strengths: The approach is well motivated from a long history of similar ideas and seems to perform quite well in practice. The authors present easy-to-use ideas for constructing intermediate distributions while admitting that these can be improved.

Weaknesses: The main weakness is that the authors never reveal which energy-based model they use in the experiments section. This would make it hard to reproduce their work.

Correctness: The claims made in the paper are founded on a theoretical framework that has found numerous applications. Empirically, the authors show that their approach is indeed superior to "naive" density ratio estimation.

Clarity: The paper is very well-written and easy to follow.

Relation to Prior Work: The relationship to AIS could be made a bit more prominent but otherwise the history of ideas with respect to bridge distributions is clearly laid out. I am not very familiar with density ratio estimation and can therefore not judge how well the present paper fits into that body of work.

Reproducibility: Yes

Additional Feedback: 1) What EBM did you use in your experiments?

[Author Response · NeurIPS 2020]

**Reviewers 1 + 3:**

⎪ *'Hard to see how TRE was relevant for energy-based modelling . . . [the] description of NCE was insufficient'*

Thank you for pointing this out; we will remedy this by strengthening the NCE connection from earlier on in the paper.

⎪ *'AIS connections could be better'*

We agree, and will add further discussion. However, we presently think the connection is more conceptual than technical.

⎪ *'MNIST is rather easy. Does it work on harder datasets?'*

⎪ *'Would higher dimension problems require considerably more intermediate distributions? If so, how bad is it?'*

While too preliminary to be included in the paper, we have some evidence from ongoing work showing that TRE can work on higher-dimensional image datasets using only modestly more intermediate distributions.

⎪ *'Is it easy to say if the limitations in [Stratos & McAllester] apply to your method and, if not, why not?'*

We believe the answer is no, since Stratos & McAllester prove strong limitations on "high-confidence lower bounds on mutual information" and our MI estimates are not lower bounds (nor upper bounds). Finally, we thank you for the relevant additional references.

**Reviewer 2:**

⎪ *'hard to assess if the claimed severity of the proposed "density-chasm" is valid. . . [the chasm] does not seem to hinder state-of-the-art results in many applications.'*

⎪ *'No comparisons to models other than "single ratio" estimations are given. It raises the question, why this problem was never addressed before, or if it was addressed before why there is no comparison.'*

As noted by reviewers 1 & 3, the "density-chasm problem" is a frequently occurring issue for practitioners using density-ratio estimation. Despite this fact, we are not aware of any prior work has clearly labelled the issue and provided a general-purpose solution. This gap in the literature motivated our paper in the first place.

In specific applications, it is often possible to design ratio-estimation tasks that avoid the density-chasm problem. As we show in our MNIST experiments, single-ratio estimation (i.e. NCE) can work very well if the noise distribution is sufficiently close to the data distribution, as is the case for the RQ-NSF model. A similar strategy (of learning a powerful noise distribution) has been used many times in the literature, and could be viewed as one of the core motivations behind GANs. However, learning *both* a ratio-estimator and complex noise distribution, in order to reduce the chasm, can be challenging/impractical, and hence it is useful for practitioners to have another method at their disposal which can work with simple noise distributions (as illustrated on MNIST in Figure 5).

The strategy of learning a good noise distribution only applies in the context of energy-based modelling. It does not apply to the more general problem setting of estimating a density-ratio $p/q$, where $p$ and $q$ are *fixed* in advance. In this general setting (which includes our MI & representation learning experiments), very few viable approaches exist. We think that the single density-ratio baselines we compare to—which include results from a 2019 Neurips paper—are representative of the state-of-the-art.

⎪ *'With little analytical guarantees and only few experiments (most of them involving variants of Gaussian noise) it is hard to assess how generalizable the results are.'*

The correctness of TRE is straightforwardly derived from that of single-density ratio estimation, for which there are extensive analytical guarantees (which we cite in Sec 2). Experiments in 4.1 & 4.2 are intentionally simple illustrations of the method using synthetic Gaussian data. Experiments in 4.3 & 4.4 use significantly more complex, non-Gaussian data. Whilst the absolute number of datasets used is modest, we believe the substantial performance increase of TRE over single ratio methods in a diverse set of applications (MI estimation, representation learning & energy-based modelling) is strong evidence of its generalizability.

⎪ *'The introduced [waymark] mechanisms all appear very ad hoc. How did you decide between the one or another?'*

As discussed in the conclusion, we agree further research on the waymark mechanisms would be valuable despite the good empirical performance of those presented, which were motivated by simplicity. As you noted, across all applications of TRE we used 'dim-wise mixing' for discrete data and the 'linear combinations' for continuous data.

⎪ *'What did you use as free parameter θ (line 88 and Fig.1)? ... [in Fig.1] The axes scaling/cut makes p(x) invisible... Can you elaborate on [line 133]? In Fig. 1(a) p(x) has sigma=1e-6 and q(x) has sigma=1 ... Could you provide a derivation of Eq. (5) from equal variances (e.g. in the appendix)?'*

$\theta$ is given in Sec 3, Eq 7 of the Appendix; we will clarify this. p(x) is missing in Fig.1 because it perfectly overlaps with the blue curve; we will amend the caption. Line 133 is slightly wrong: all of our experiments *except* 4.1 (i.e. Fig 1) preserve variance; we will amend it and add a one-line derivation of the variance property. Thank you for the comments.

**Reviewer 4**

⎪ *'The main weakness is that the authors never reveal which energy-based model they use in the experiments section.'*

We state in Sec 3.4 that the model has the form $r(\mathbf{x}; \boldsymbol{\theta})q(\mathbf{x})$, where $r(\mathbf{x}; \boldsymbol{\theta})$ is the product of all the bridges (Eq. 4) and $q(\mathbf{x})$ is a noise distribution. In Sec 4.4, we state how each bridge is parameterised, and the different choices of noise distribution. Sec 1 & 5 of the appendix give all the architectural/training details necessary to reproduce our results.

⎪ *'The relationship to AIS could be made a bit more prominent'*

We agree, and will include more discussion in the final paper (please see also our response to Revs 1+3).

[Meta-Review · NeurIPS 2020]

All three reviewers agree that this paper presents a novel solution with great relevance for NeurIPS; this is a clear accept.